# A Benchmark and Pair-Level 4PL-IRT Framework for Reliable Evaluation of LLM Reasoning

## 1 Abstract

### Abstract

Large Language Models (LLMs) have demonstrated remarkable capabilities across a wide range of tasks, yet reliably evaluating their reasoning ability, particularly in symbolic reasoning, remains an open challenge. In this work, we introduce a novel evaluation framework based on Item Response Theory (IRT), applied at both the *pair level* and *instance level*, and compare its effectiveness against traditional metrics such as Accuracy, F1, and MCC. Through extensive experiments across multiple LLMs, we show that while conventional metrics provide limited and sometimes misleading signals, IRT-based measures—especially under the 4PL model at the pair level—offer more stable and reliable insights into the reasoning competence of LLMs. Our study further presents a new benchmark suite for symbolic reasoning, along with a principled methodology for its generation and evaluation. This framework not only highlights the shortcomings of standard metrics, but also establishes IRT as a more trustworthy foundation for assessing the reasoning abilities of LLMs. We argue that such rigorous evaluation methods are essential for guiding the future development of LLMs toward robust reasoning performance.

## 2 Introduction

## 3 Apply IRT on 3 ways checking

Figure 6 presents a comparison of the estimated latent abilities ($\theta$) of different models under the 2PL, 3PL, and 4PL item response theory (IRT) frameworks. Several patterns emerge:

1. **GPT-5 consistently achieves the highest ability estimates** across all three models, confirming its superior performance in the ADR-Strict task.

2. Under the **2PL model**, most systems fall within a moderate negative range ($-15 \leq \theta \leq 12$), reflecting average-to-low performance.

3. The **3PL model penalizes models that appear to rely on guessing**, which drives their estimated abilities substantially lower (e.g., the *gpt-3.5* family and *gpt-4-turbo*).

4. By contrast, the **4PL model introduces an upper asymptote**, which allows models with partial but inconsistent success (e.g., *o1* and *deepseek-reasoner*) to receive more favorable ability estimates compared to 2PL and 3PL.

5. Overall, the contrast between models highlights how the choice of IRT formulation (2PL vs. 3PL vs. 4PL) affects the interpretation of model "ability": 2PL emphasizes overall average skill, 3PL heavily discounts potential guessing behavior, while 4PL can capture high but uneven performance by lifting ability estimates for models with stronger peaks.

These results suggest that **IRT not only provides a relative ranking of LLMs on the evaluated task but also offers theoretical insight into their response patterns**—whether their success comes

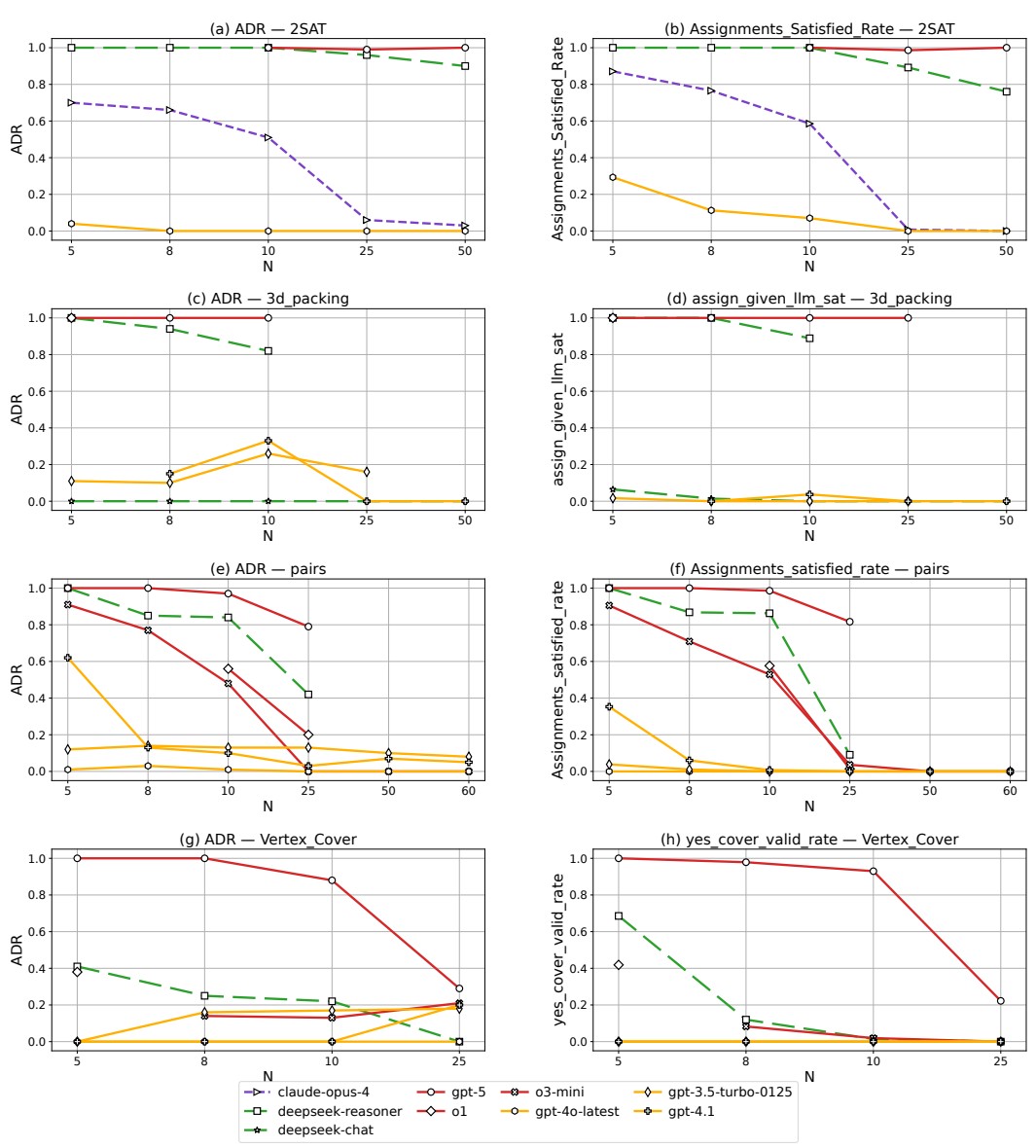

Figure 1: Comparison of ADR and task-specific metrics across the four problem settings: 2SAT, pairs, 3D packing, and Vertex Cover.

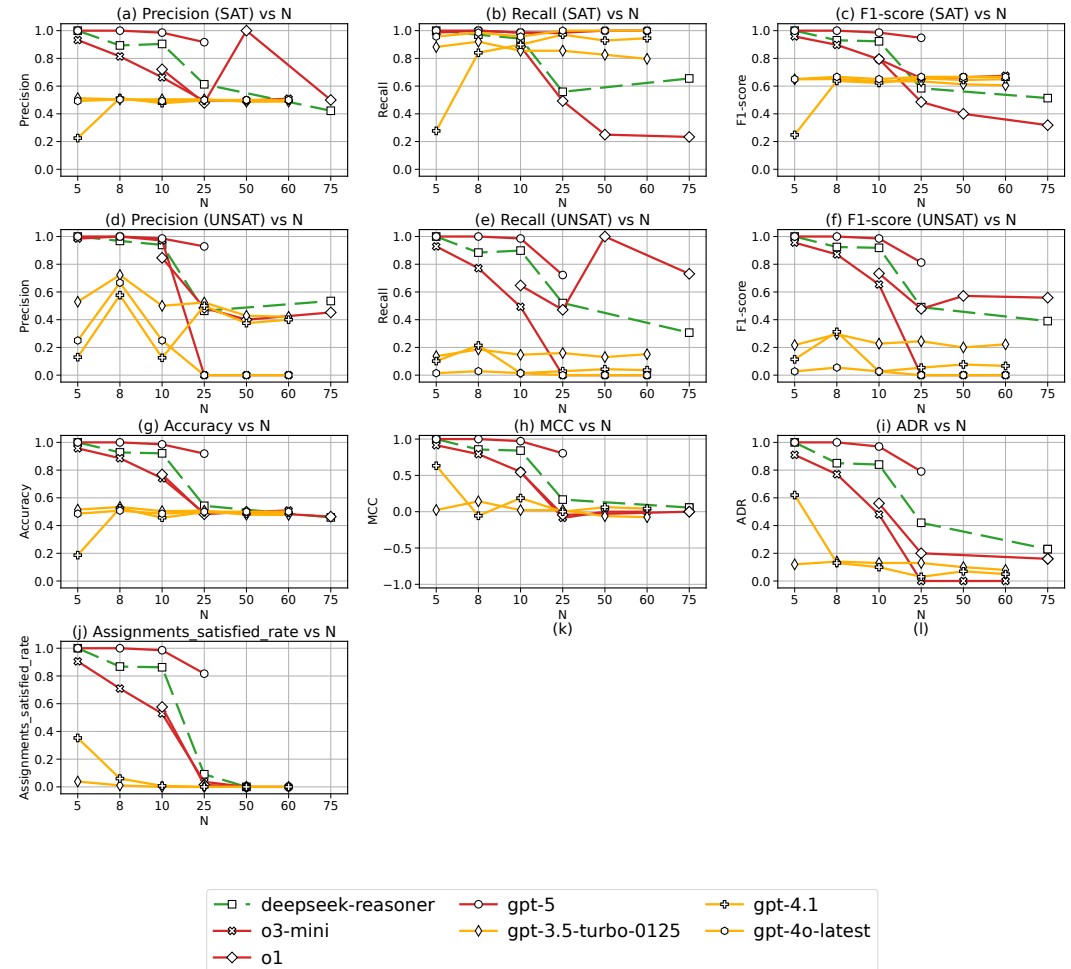

Figure 2: Prediction accuracy on paired instances (small $\alpha$).

from consistent competence (2PL), inflated by guessing (3PL), or bounded by task-specific asymptotic performance (4PL).

**Observations.** Figure 9 illustrates how item parameters distribute under three IRT models:

- **2PL (Fig. 9a)**: Discrimination values ($a$) are overall small and tightly clustered; item difficulties ($b$) are concentrated around zero. Since 2PL models only $(a, b)$, it implicitly assumes no guessing and a perfect upper asymptote, which compresses $a$ estimates at the extremes.

- **3PL (Fig. 9b)**: $a$ values are higher and more separable. By explicitly modeling the *guessing parameter $c$*, the model isolates "random hits" from true discrimination. This makes $a$ more sensitive to ability differences.

- **4PL (Fig. 9c)**: Maintains stable $a$ values across a wide $b$ range, with structured clusters aligned with different $N$. The extra *upper asymptote $d$* prevents easy items from forcing $a$ downward, leading to more stable estimates across conditions.

**Model forms.** For subject ability $\theta$, the probability of a correct response is:

$$\text{2PL:} \quad P(Y{=}1 \mid \theta) \;=\; \sigma\big(a(\theta - b)\big), \tag{1}$$

$$\text{3PL:} \quad P(Y{=}1 \mid \theta) \;=\; c + (1 - c)\,\sigma\big(a(\theta - b)\big), \tag{2}$$

$$\text{4PL:} \quad P(Y{=}1 \mid \theta) \;=\; c + (d - c)\,\sigma\big(a(\theta - b)\big), \tag{3}$$

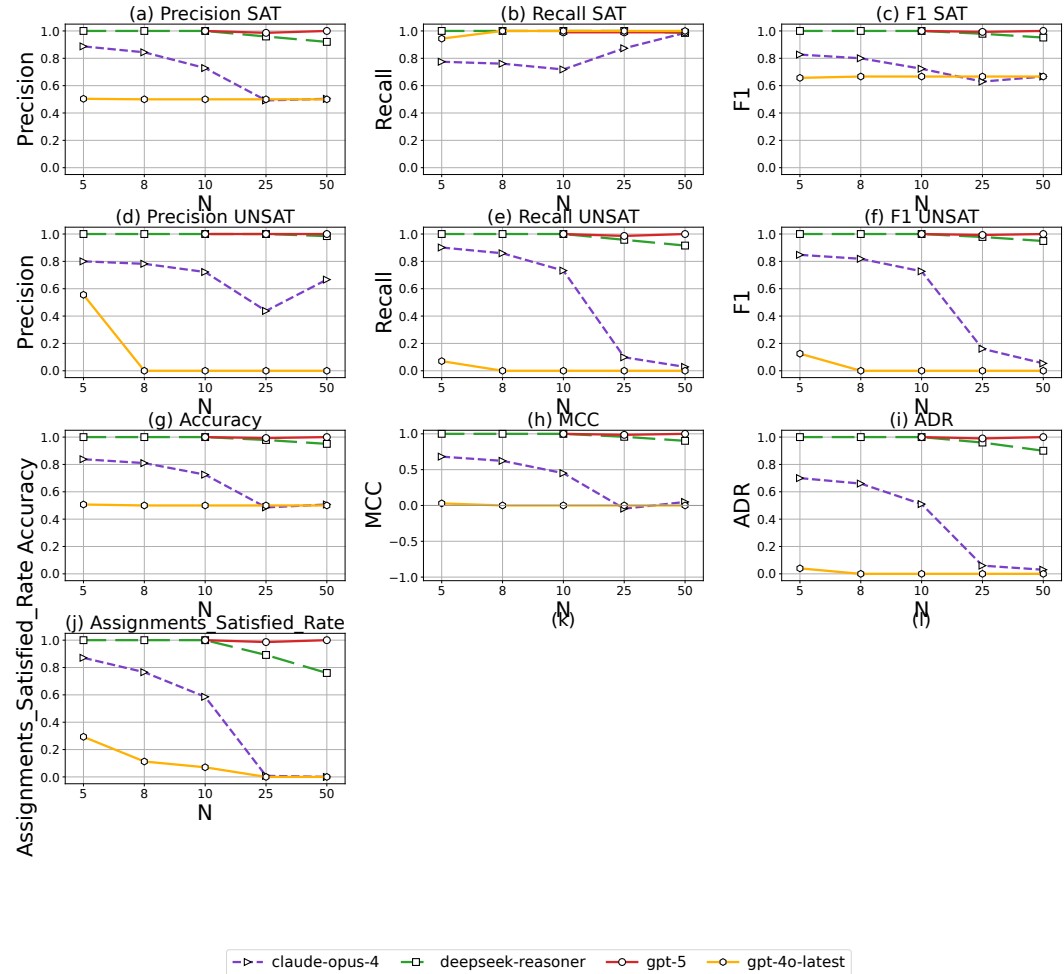

Figure 3: Overall results on the 2SAT benchmarks.

where $\sigma(x) = 1/(1 + e^{-x})$, $a$ is discrimination, $b$ is difficulty, $c$ is the lower asymptote (guessing), and $d$ is the upper asymptote (ceiling).

**Advantages and limitations.**

- **2PL**
  - *Pros:* Simple, stable, easy to estimate and interpret. Works well for medium-difficulty items.
  - *Cons:* Cannot capture guessing or ceiling effects; biased at the extremes.
- **3PL**
  - *Pros:* Explicitly models guessing ($c$), making discrimination $a$ more interpretable in tasks where random success is non-negligible.
  - *Cons:* Adds an extra parameter, risking identifiability and higher variance when data are limited.
- **4PL**
  - *Pros:* Models both guessing ($c$) and ceiling effects ($d$), giving realistic asymptotes at both ends. Provides more stable estimates of $a, b$ across different $N$, as seen in Fig. 9c.
  - *Cons:* Most complex, requires more data and regularization to ensure robust parameter estimates.

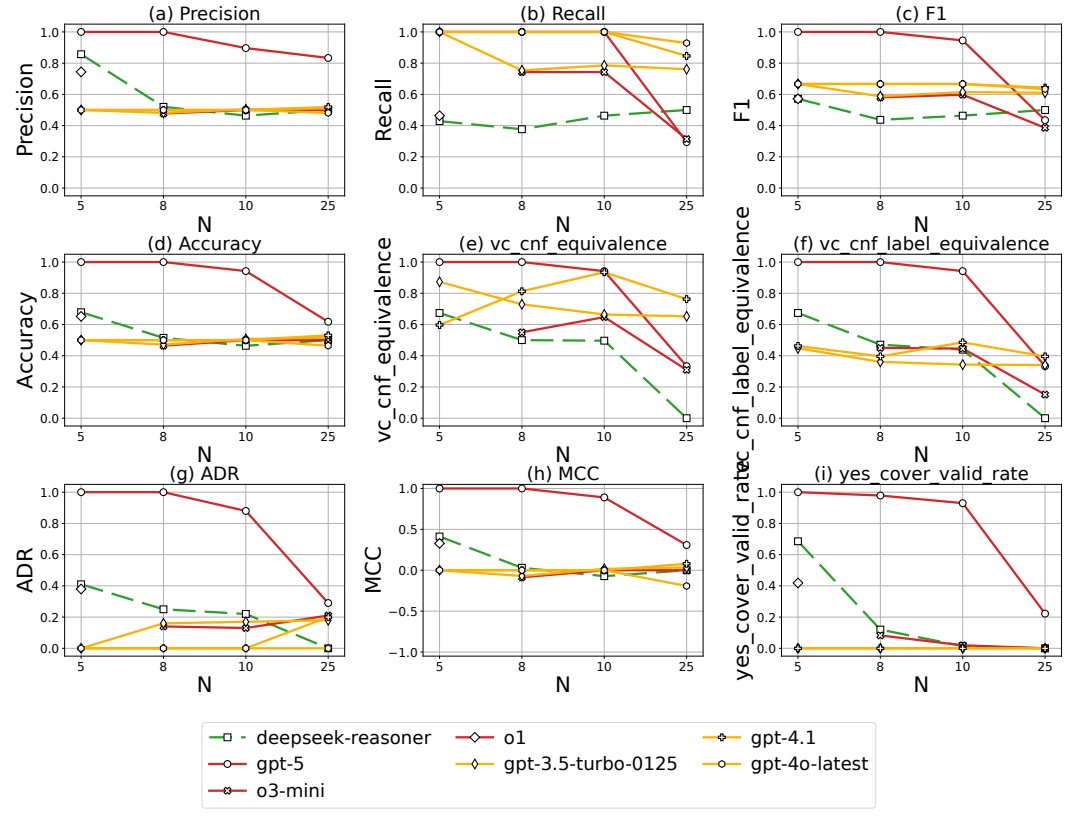

Figure 4: Summary of Vertex Cover metrics across models.

**Comparison with traditional metrics.** Conventional metrics such as *accuracy*, *CTT item difficulty/discrimination*, and *ROC-AUC* lack explicit item modeling:

- **Accuracy**: highly dataset-dependent; cannot explain *why* performance differs.
- **CTT indices**: sensitive to sample distribution; cannot handle guessing or ceiling effects.
- **ROC-AUC**: useful as a global rank metric but not item-level, and does not separate $c$ or $d$ effects.

By contrast, **3PL** isolates guessing effects, while **4PL** additionally captures ceiling effects. This explains why the 4PL model produces more consistent item parameter structures across varying $N$, as evident in Fig. 9c.

**Conclusion.** When datasets involve *non-negligible guessing* or *imperfect ceiling effects*, **3PL and especially 4PL are recommended**. The 4PL model is particularly advantageous in complex real-world scenarios, offering robust and interpretable item parameters that remain stable across conditions.

## 4 APPLY IRT ON PAIR LEVEL

## 5 APPLY IRT ON INSTANCE LEVEL

## 6 TRADITION METRICS

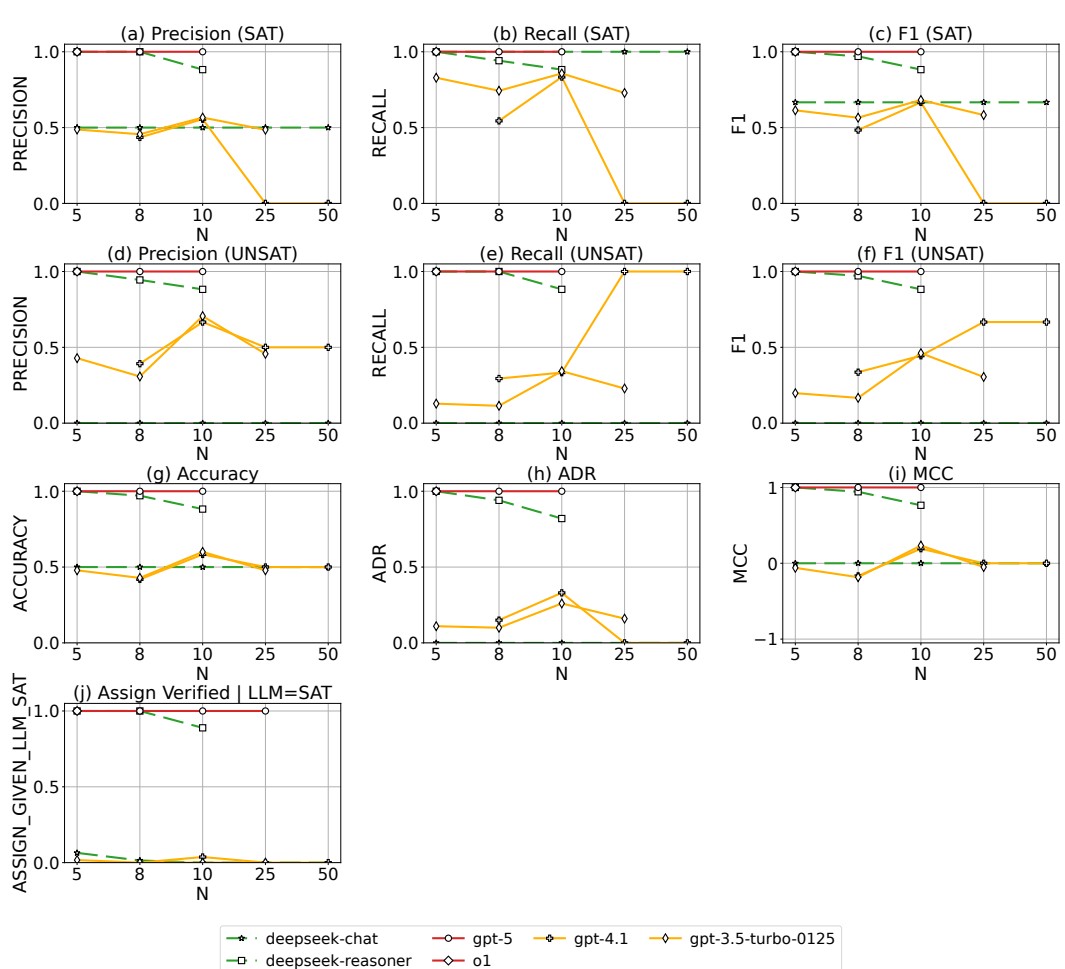

Figure 5: 3D packing results: LLM prediction metrics (3×3 and 4×3 panels).

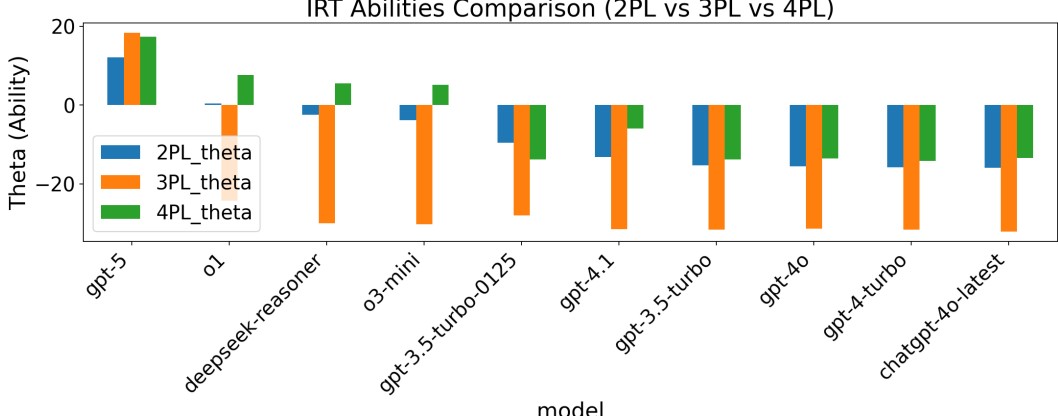

Figure 6: 3 ways checking: Comparison of estimated latent abilities ($\theta$) of different models under the 2PL, 3PL, and 4PL IRT frameworks.

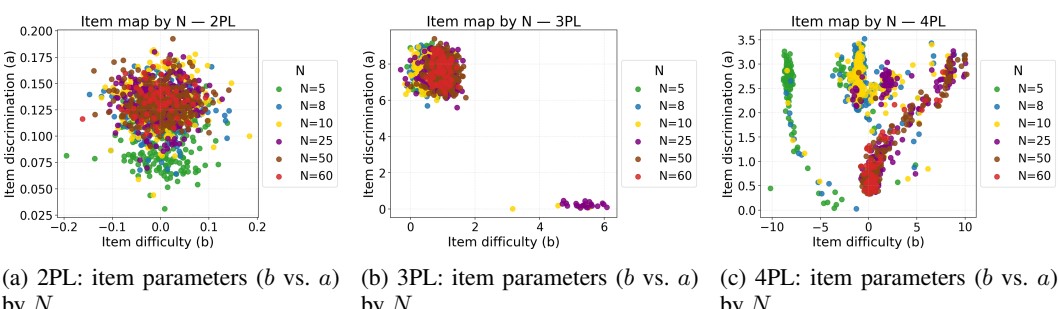

(a) 2PL: item parameters ($b$ vs. $a$) by $N$

(b) 3PL: item parameters ($b$ vs. $a$) by $N$

(c) 4PL: item parameters ($b$ vs. $a$) by $N$

Figure 7: **3 ways checking: Item parameter distributions (difficulty $b$ vs. discrimination $a$), with colors indicating pair size $N$.** Color mapping: green = $N$=5, blue = $N$=8, yellow = $N$=10, purple = $N$=25, brown = $N$=50, red = $N$=60. The clustering of colored point clouds across $(b, a)$ space reflects how items with different $N$ values vary in difficulty and discrimination.

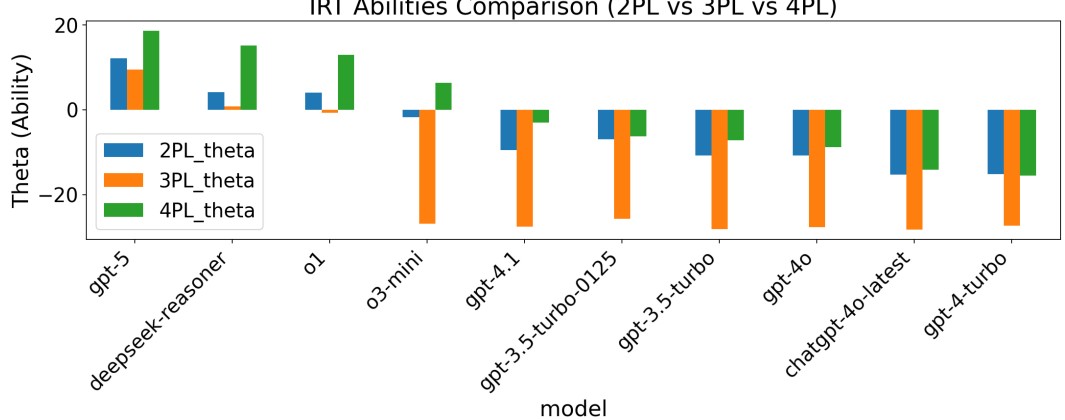

Figure 8: Pair Level: IRT Abilities Comparison across models

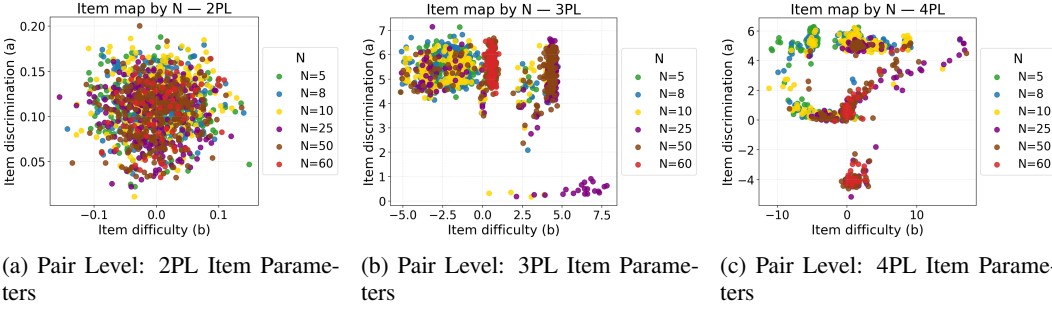

(a) Pair Level: 2PL Item Parameters

(b) Pair Level: 3PL Item Parameters

(c) Pair Level: 4PL Item Parameters

Figure 9: Pair Level: Item parameter maps for 2PL, 3PL, and 4PL models (by N).

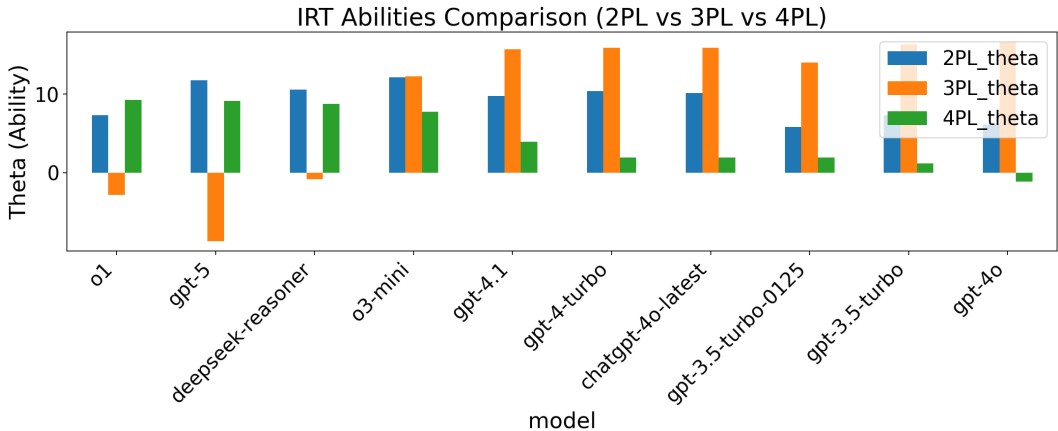

Figure 10: Instance-level: Comparison of model abilities estimated by 2PL, 3PL, and 4PL.

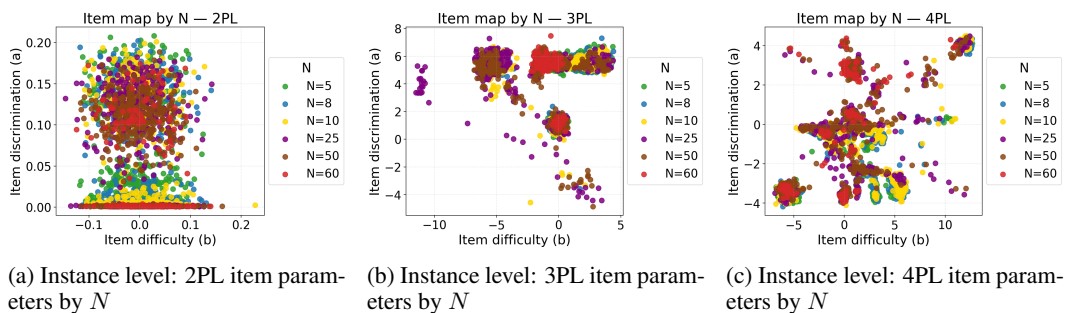

(a) Instance level: 2PL item parameters by $N$

(b) Instance level: 3PL item parameters by $N$

(c) Instance level: 4PL item parameters by $N$

Figure 11: Instance-level: Item discrimination ($a$) versus difficulty ($b$) under different IRT models.

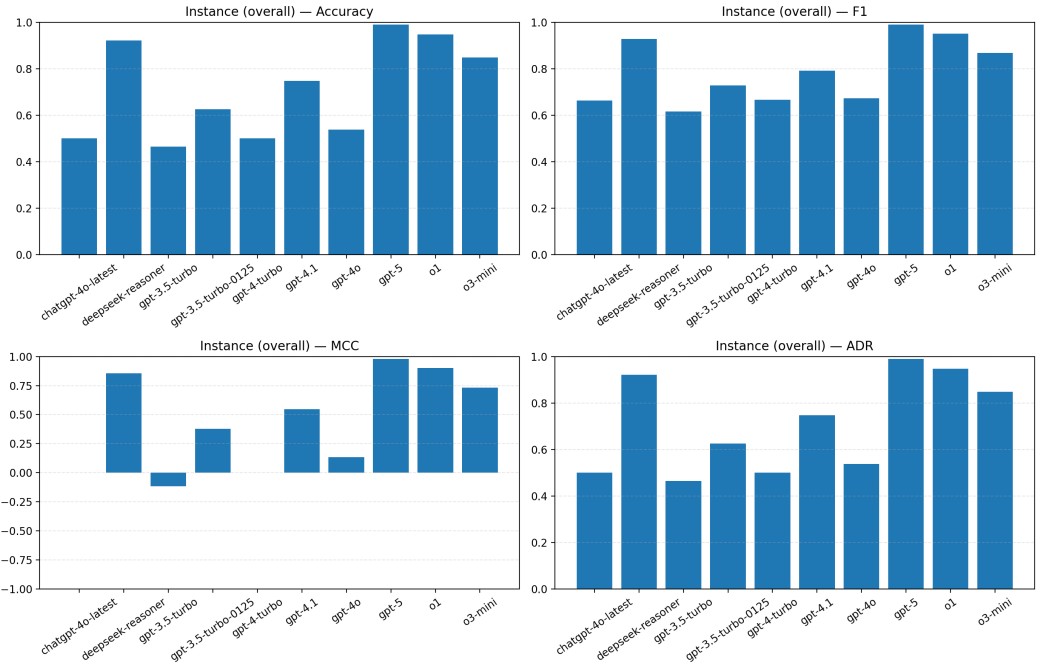

Figure 12: Overall instance-level metrics (2×2).

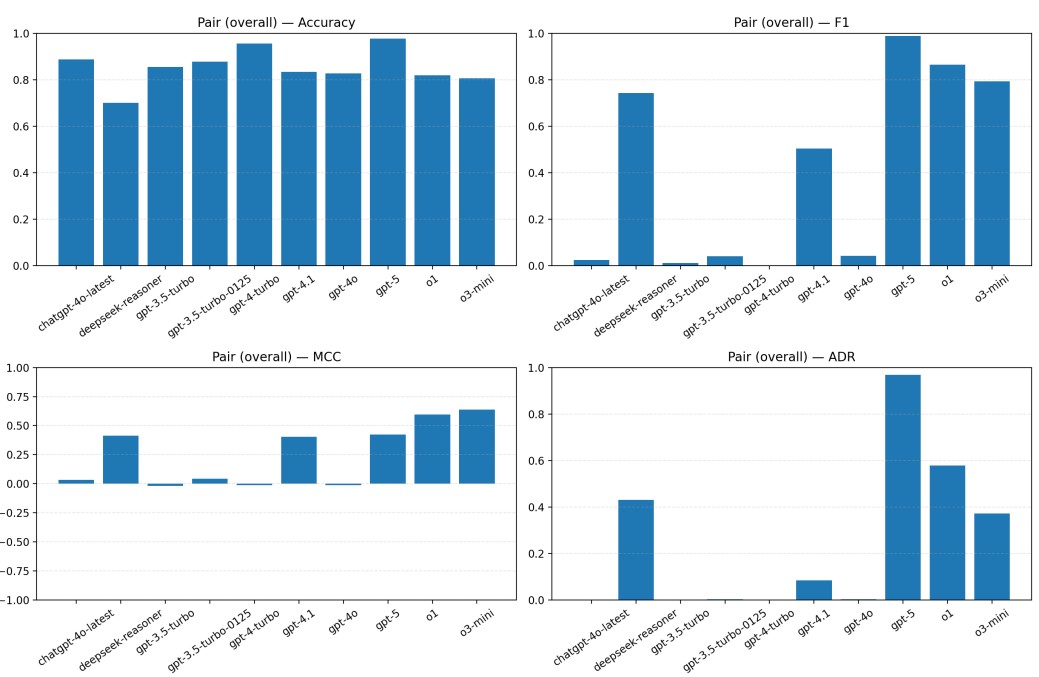

Figure 13: Overall pair-level metrics (2×2).

