# OpenReview forum: "A Benchmark and Pair-Level 4PL-IRT Framework for Reliable Evaluation of LLM Reasoning"
_ICLR.cc/2026/Conference — Submitted to ICLR 2026_

### Official Review · Reviewer_eh42 · 2025-10-26

**Soundness:** 3
**Presentation:** 3
**Contribution:** 3
**Rating:** 6
**Confidence:** 4

**Summary:**

The paper argues that standard metrics for evaluating Large Language Models (LLMs)—such as Accuracy, F1-score, and MCC—are unreliable and often misleading, especially for complex symbolic reasoning tasks. To solve this, the authors propose a new evaluation framework based on Item Response Theory (IRT), a sophisticated statistical model used in psychometrics (e.g., standardized testing) to measure a subject's "latent ability" (their true skill level). The authors conclude that the 4PL-IRT model is the most stable and reliable method. It avoids the pitfalls of simple accuracy and, by accounting for both guessing and ceiling effects, provides a much more trustworthy assessment of an LLM's true reasoning competence. The paper introduces a new symbolic reasoning benchmark to test this framework.

**Strengths:**

1. its direct and well-supported critique of using simple accuracy to measure reasoning. It clearly shows (in Figures 1-5) how metrics like Accuracy and F1 are unstable and fail to capture the nuances of a model's ability.

2. applying IRT to LLM evaluation is a novel and powerful idea. Instead of just a simple score, IRT provides a model-based estimate of a model's "ability" that is separate from the "difficulty"  of the questions, which is far more insightful.

3. the comparison of 2PL, 3PL, and 4PL models makes a compelling case for the 4PL model, as it uniquely accounts for both random "guessing" at the low end and "ceiling effects" (inconsistent success) at the high end. This provides a much more realistic picture of a model's behavior.

4. it contributes a new benchmark suite specifically for symbolic reasoning (2SAT, 3D packing, Vertex Cover), which is essential for rigorously testing its new evaluation framework.

**Weaknesses:**

1. the proposed 4PL-IRT framework is vastly more complex to implement and interpret than simply calculating accuracy. This presents a significant barrier to adoption for many researchers and developers who may lack the necessary psychometric background.

2. more complex statistical models like 3PL and 4PL require more data to produce stable and reliable parameter estimates. The paper notes this (lines 204-209), suggesting this framework may be unsuitable for small-scale benchmarks or quick evaluations.

**Questions:**

Is the complexity of the 4PL-IRT model truly necessary? Could simpler, more interpretable metrics (perhaps improved versions of traditional ones or specific behavioral tests) achieve similar reliability without the high statistical overhead?

---

### Official Review · Reviewer_bfV1 · 2025-10-29

**Soundness:** 1
**Presentation:** 1
**Contribution:** 1
**Rating:** 0
**Confidence:** 1

**Summary:**

This paper intends to introduce a new evaluation framework for symbolic reasoning based on Item Response Theory (IRT). However, the paper appears to be incomplete, missing substantial sections that describe the proposed framework and the experiments conducted. Given that only one of the six chapters contains text (in bullet points), it is not possible to properly assess its content or contributions.

**Strengths:**

N/A — Given the current state of the paper, it is not possible to assess the strength of its contributions.

This should have been a desk reject IMO.

**Weaknesses:**

As mentioned before, the paper appears to be incomplete. Since substantial parts are missing and its content and contributions cannot be properly assessed, as such and mentioned above I suggest (desk-)rejecting this paper.

**Questions:**

n/a

---

### Official Review · Reviewer_8H81 · 2025-11-01

**Soundness:** 2
**Presentation:** 2
**Contribution:** 1
**Rating:** 2
**Confidence:** 4

**Summary:**

The paper introduces a novel framework for evaluating the reasoning abilities of large language models (LLMs) using Item Response Theory (IRT), with a focus on symbolic reasoning tasks. The authors propose a pair-level 4PL-IRT model for more reliable and interpretable assessment of model performance. They compare this new framework against traditional metrics (such as accuracy, F1, and MCC) and demonstrate that IRT-based evaluations, especially at the pair level, provide more stable and insightful analysis. Through extensive experiments, the paper shows how different IRT models (2PL, 3PL, 4PL) capture varying aspects of model performance, offering more nuanced insights into LLM capabilities.

**Strengths:**

The authors provide a thorough explanation of the IRT models (2PL, 3PL, and 4PL) and how they differ in their ability to capture various aspects of LLM performance. The comparison across these models is well-detailed, highlighting the benefits of the 4PL model in dealing with guessing and ceiling effects.

**Weaknesses:**

1.While the paper focuses on IRT models, it could benefit from further exploration of other evaluation frameworks or hybrid models that combine traditional metrics with IRT for more comprehensive insights. The focus on symbolic reasoning tasks could also be expanded to include other domains.

2.The paper does not address the computational overhead of using IRT, especially the more complex 4PL model. This could be a concern when applying the methodology to large-scale LLMs or tasks with a high volume of data. Further discussion on the scalability of the approach would enhance its practical value.

3.The experiments focus heavily on comparing models within specific tasks, but there is limited exploration of how generalizable the IRT framework is across diverse problem domains. It would be useful to see how well this approach extends to tasks outside symbolic reasoning, such as natural language understanding.

4.The paper would greatly benefit from a framework diagram to clearly illustrate the proposed methodology. While the experiments are comprehensive, the analysis of the experimental results could be more detailed. The paper presents a lot of data but doesn’t provide an in-depth breakdown of how the results vary across models and tasks. A more qualitative analysis discussing the significance of the observed patterns, and a deeper dive into the implications of these results for model development, would enhance the paper’s contribution.

**Questions:**

please see weakness.

---

### Official Review · Reviewer_XcMH · 2025-11-02

**Soundness:** 1
**Presentation:** 1
**Contribution:** 1
**Rating:** 0
**Confidence:** 4

**Summary:**

I think a crucial mistake was made by the authors when they composed the manuscript - Many paragraphs of many sections are missing.

**Strengths:**

In complete submission.

**Weaknesses:**

Incomplete submission.

**Questions:**

Is it only me that failed to see so many paragraphs in the pdf?

---

### Meta-Review · Area_Chair_etc7 · 2026-01-06

**Summary:**

The reviewers raised serious concerns about the submission's completeness: multiple reviewers noted that large sections of the paper are missing, rendering the proposed framework and experiments unassessable. While one reviewer found the IRT-based evaluation idea promising, others could not evaluate contributions due to the incomplete manuscript. Given the lack of essential content—including methodology, results, and analysis—the submission fails to meet basic standards for review. This warrants a rejection, as the paper in its current state does not constitute a complete scientific contribution.

**Reviewer Concerns:**

Since there is no rebuttal, no concerns are addressed.
Concerns about missing parts, expression and experiments are still outstanding.

**Reviewer Scores:**

No reviewer may have changed the score(s).
So the scores may be 0,2,0,6. I think the scores are below the acceptance threshold.

---

### Decision · Program_Chairs · 2026-01-26

Reject